# Minocycline as Treatment for Psychiatric and Neurological Conditions: A Systematic Review and Meta-Analysis

**DOI:** 10.3390/ijms24065250

**Published:** 2023-03-09

**Authors:** Bruna Panizzutti, David Skvarc, Sylvia Lin, Sarah Croce, Alcy Meehan, Chiara Cristina Bortolasci, Wolfgang Marx, Adam J. Walker, Kyoko Hasebe, Bianca E. Kavanagh, Margaret J. Morris, Mohammadreza Mohebbi, Alyna Turner, Laura Gray, Lesley Berk, Ken Walder, Michael Berk, Olivia M. Dean

**Affiliations:** 1IMPACT, Institute for Innovation in Physical and Mental Health and Clinical Translation, Barwon Health, School of Medicine, Deakin University, Geelong, VIC 3220, Australia; 2School of Psychology, Faculty of Health, Deakin University, Geelong, VIC 3220, Australia; 3Melbourne Neuropsychiatry Centre, University of Melbourne, Parkville, VIC 3053, Australia; 4School of Biomedical Sciences, UNSW Sydney, Kensington, NSW 2052, Australia; 5Biostatistics Unit, Faculty of Health, Deakin University, Burwood, VIC 3125, Australia; 6Florey Institute for Neuroscience and Mental Health, University of Melbourne, Parkville, VIC 3052, Australia; 7Orygen, The National Centre of Excellence in Youth Mental Health, Parkville, VIC 3052, Australia

**Keywords:** minocycline, psychiatry, neurology, adjunctive treatment, meta-analysis

## Abstract

Minocycline has anti-inflammatory, antioxidant, and anti-apoptotic properties that explain the renewed interest in its use as an adjunctive treatment for psychiatric and neurological conditions. Following the completion of several new clinical trials using minocycline, we proposed an up-to-date systematic review and meta-analysis of the data available. The PICO (patient/population, intervention, comparison and outcomes) framework was used to search 5 databases aiming to identify randomized controlled trials that used minocycline as an adjunctive treatment for psychiatric and neurological conditions. Search results, data extraction, and risk of bias were performed by two independent authors for each publication. Quantitative meta-analysis was performed using RevMan software. Literature search and review resulted in 32 studies being included in this review: 10 in schizophrenia, 3 studies in depression, and 7 in stroke, with the benefit of minocycline being used in some of the core symptoms evaluated; 2 in bipolar disorder and 2 in substance use, without demonstrating a benefit for using minocycline; 1 in obsessive-compulsive disorder, 2 in brain and spinal injuries, 2 in amyotrophic lateral sclerosis, 1 in Alzheimer’s disease, 1 in multiple systems atrophy, and 1 in pain, with mixes results. For most of the conditions included in this review the data is still limited and difficult to interpret, warranting more well-designed and powered studies. On the other hand, the studies available for schizophrenia seem to suggest an overall benefit favoring the use of minocycline as an adjunctive treatment.

## 1. Introduction

Minocycline is a tetracycline antibiotic traditionally used acutely to treat bacterial infections and in longer-term use to treat acne [1]. In addition to its antimicrobial activity, minocycline has a range of other mechanisms of action that may be relevant for non-infectious disorders. This explains the renewed interest in repurposing minocycline for new applications. The approach for using minocycline in psychiatric and neurological conditions leverages the ‘new’ knowledge which associates its anti-inflammatory effects with the abnormalities presented in these conditions. In a similar way that ixekizumab was repurposed as an antidepressant in 2016 [2]. This systematic review will explore the current use of minocycline for a range of psychiatric and neurological disorders. Critical evaluation of the available evidence will better inform clinicians’ decisions regarding the use of minocycline in patient treatment regimens [1,3]. This review is also timely in the context of adjunctive anti-inflammatory treatments where there is often heterogeneity in the literature regarding the field broadly and specific agents being trialed. Therapeutic mechanisms of minocycline for non-infectious disorders.

Minocycline has been shown to inhibit microglial activation following brain inflammation, a mechanism of clear relevance to both psychiatric and neurological conditions where increased inflammation has been implicated in the pathogenesis across several neuropsychiatric disorders [1]. This is in parallel with direct effects on inflammatory markers [1], where it has also been shown to alter a range of inflammatory markers including cytokines such as tumor necrosis factor-alpha (TNF-α) [4]. Furthermore, minocycline reduces oxidative stress and improves antioxidant function, supporting neuronal repair and function. Minocycline also promotes neurogenesis via augmentation of brain-derived neurotrophic factor (BDNF) [1]. 

In regards to the clinical investigation of minocycline for psychiatric disorders, two previous systematic reviews have explored minocycline as a treatment for major mental disorders [5,6]. These reviews, published in 2014 and 2019, reported that there was sufficient evidence to show that adjunctive minocycline treatment was associated with improved symptoms of schizophrenia but there was insufficient evidence (lack of studies) for other disorders. This has also been shown in several other systematic reviews exploring schizophrenia and psychotic disorders, specifically [7,8,9,10,11,12]. Finally, a systematic review reported limited support for the use of adjunctive minocycline treatment for obsessive-compulsive disorder [13]. 

Regarding the use of minocycline for neurological disorders, most previous reviews exploring the use of minocycline for neurological conditions focused on stroke or traumatic brain injury. When exploring minocycline treatment for stroke, minocycline was found to reduce brain-associated inflammation [14,15,16]. This translated into better clinical outcomes for people who had experienced a stroke. Similar findings are also reported in traumatic brain injury [17,18,19,20]. Given the overlap in pathophysiology, especially the elevated inflammation in both conditions, the similarities in findings are unsurprising. Both the stroke and traumatic brain injury literature are hampered by small sample sizes and heterogeneous study designs, making overall definitive conclusions difficult. Two recent reviews were published on neuropathic pain, but neither of these applied a completely systematic approach [21,22]. The findings from these two reviews suggest minocycline may be useful for controlling certain types of pain (e.g., diabetic and leprotic neuropathies) but not others (such as lumbar radicular pain). These conclusions are limited by the number of studies and the heterogeneity of pain disorders [21,22]. 

While several systematic reviews have already been conducted for minocycline in psychiatry over the last decade, following the completion of a number of new trials, an update in this field is warranted. Hence, this systematic review includes the evaluation of all clinical trial data available from inclusion to June 2022 regarding the use of minocycline in psychiatric and neurological disorders. A meta-analysis has been conducted, where statistically appropriate, to provide objective measurement of the clinical effects.

## 2. Methods

This systematic review was developed in accordance with the PRISMA reporting guidelines [23]. A protocol detailing the methods planned for this review has been prospectively published [24]. 

### 2.1. Search Strategy

The PICO (Patient/Problem/Population, Intervention, Comparison/Control, Outcome) framework was used to develop the search strategy for this systematic review aiming to identify randomized controlled trials of minocycline for the treatment of core symptoms of psychiatric and neurological conditions. A systematic online literature search of PubMed, Embase, Cochrane, CINAHL (through EBSCO), and PsycINFO (through EBSCO), from inception through to May 2022 was conducted using the search strategy in Table 1. The studies identified during the search were then imported to COVIDENCE [25] for data management and screening. For other search strategies see Appendix A. 

### 2.2. Study Selection

One author (BP) applied the search to the selected databases and imported the identified studies to Covidence. The screening process was conducted independently by two authors (BP and CCB) with a third author acting as adjudicator (KH) in case of discordance. The title and abstracts were first screened for eligibility using the following inclusion criteria: (a) randomized clinical trial, (b) use of minocycline, (c) any size and duration, (d) adult population (≥18 years), and (e) placebo or active—controlled. Studies were excluded if they were a (a) pre-clinical study, (b) clinical trial protocol, (c) not population of interest, and (d) a safety and tolerability study. The same reviewers then assessed the studies’ full text and those that met all inclusion criteria were selected for data extraction.

### 2.3. Data Extraction

Data extraction was performed by five of the authors (BP, AM, DD, SC, and SL), ensuring each paper was reviewed and data extracted by two authors. Data were extracted independently, and discrepancies were resolved via discussion or through an adjudicator (OMD). The revised version of the Cochrane Collaboration’s Risk of Bias tool [26,27] was used to assess the risk of bias in the randomized studies.

### 2.4. Quality Assessment

The risk of bias was assessed independently by the researchers who performed the data extraction (BP, AM, DD, SC, and SL) using the Cochrane risk of bias tool [28]. The following domains were assessed: selection bias (random sequence generation, allocation concealment), detection bias (blinding of outcome assessment), attrition bias (incomplete outcome data), reporting bias (selective reporting), and other bias (sample selection bias) [29,30,31,32], contamination bias [30,31], compliance bias [31] response bias [33], and performance bias (blinding of participants). Under each domain, studies were classified as low, high, or unclear risk of bias. Discrepancies were resolved through discussion between the two researchers.

### 2.5. Data Analysis

RevMan software version 5.4.1 [34] (Review Manager (RevMan) [Computer program], 2020) was used to perform a quantitative meta-analysis. Inverse-variance random effect models with restricted maximum likelihood estimation (REML) were used to calculate standard effect sizes for both individual measures within cognitive domains, and the summary effect size for each domain overall. All effect sizes represent the standardized mean difference (SMD) in continuous outcomes between minocycline and comparator groups after the intervention period. Risk ratios were calculated for categorical outcomes. The heterogeneity of effect was calculated as I2, where increasing values represent increasing heterogeneity between studies, and values of 25%, 50%, and 75% represent approximately low, moderate, and high levels of heterogeneity, respectively [35]. Where heterogeneity was high, we performed meta-regression to examine potential moderators of the effects using participant age, intervention length, baseline outcome scores, and risk of bias, where at least ten studies were present. For outcomes with high heterogeneity and fewer than ten studies, a one-in-one-out analysis was performed to identify sources of heterogeneity and test the robustness of the results. Publication bias was examined using a funnel plot, Kendall’s tau for continuous outcomes, and Egger’s regression test for categorical outcomes. 

Studies examining psychiatric disorders have been examined in two ways: results related to clinical symptoms (e.g., depressive symptoms) were grouped according to primary diagnosis (i.e., schizophrenia, major depressive disorder, bipolar disorder, etc.) while cognitive outcomes were grouped according to the National Institute of Mental Health’s Measurement and Treatment Research to Improve Cognition in Schizophrenia—Consensus Cognitive Battery (MATRICS-CCB) cognitive domains [36]. All outcomes were compared between groups (minocycline and controls) at follow-up. Some scores have been reversed to ensure uniformity of forest plots (i.e., all effects to the left favor minocycline). Where a single study used multiple methods of scoring a single outcome, results were pooled, and the study was noted as having a combined outcome. If a study reported heterogeneous samples that were otherwise treated identically (i.e., different countries or severity of symptoms), these samples were included as separate entries within the meta-analyses.

## 3. Results and Discussion

### 3.1. Search Yield and Study Inclusion

The PRISMA flowchart is presented in Figure 1. The search of the 5 chosen databases identified 1722 studies, of which 421 were identified as duplicates. Going forward, 1301 titles and abstracts were screened, and 1162 studies were excluded for not meeting the inclusion criteria. The full text of 139 studies were assessed and a further 71 studies were excluded. The searches and screening process described in this study resulted in the inclusion of 68 independent studies. During the data extraction, 36 studies were reconsidered and excluded for the following reasons: 7 were duplicates, 2 were futility studies, 4 were studies on safety and tolerability, 2 were pilot studies without outcome data, 9 did not present data on the improvement of core symptoms, 4 studies did not include the target population (1 in participants < 18 years old, 2 in diabetic patients, 1 on opioid dependence (recreational)), and 8 studies were sub-analyses of previous studies focusing on biological/molecular data. Thirty-two studies were analyzed regarding the effect of minocycline on the core symptoms of psychiatric and neurological conditions (Table 2).

### 3.2. Psychiatric Disorders

#### 3.2.1. Schizophrenia

Participants taking adjunctive minocycline to treat schizophrenia reduced total symptoms (−0.36 [−0.65, −0.07], I^2^ = 71%), negative symptoms (−0.69 [−1.05, −0.33]. I^2^ = 85%), and general symptoms (−0.33 [−0.64, −0.02], I^2^ = 63%); however, no significant improvement was reported for positive symptoms (−0.12 [−0.28, 0.04], I^2^ = 8%) or any cognitive domain (Figure 2). The inclusion of meta-regressive moderators reduced the heterogeneity for total symptoms to 65%. One-in-one-out analysis revealed Khodaie-Ardakani (2014) as the largest source of heterogeneity, which was reduced to 34% when the study was removed and the effect size was reduced to −0.26 [−0.52, −0.00]. Meta-regressive moderators reduced the heterogeneity for negative symptoms to 45%, and the effect size remained stable at −0.62 [−0.97, −0.27]. For general symptoms, the removal of Chaudry et al., (2012, Brazil) reduced heterogeneity to 35% and the effect became null; −0.21 [−0.44, 0.02]. We observed no evidence of publication bias (Appendix A).

Consistent with previous reviews, the disorder with the greatest representation in this systematic review is schizophrenia with 10 trials included in the current review. This represents 415 adult participants receiving minocycline. Three studies were conducted in Iran, three in China, and one each in the following countries: the UK, Pakistan, Brazil, the USA, Romania, and the Republic of Moldova. All the studies included had a final dose of 200 mg of adjunctive minocycline daily compared to a placebo group. Treatment time varied from 8 weeks to 12 months. 

More specifically, the BeneMin [44] trial included participants on their first episode of schizophrenia or schizoaffective psychosis and used adjunctive minocycline treatment as usual for 12 months and a follow-up visit at 15 months. This study followed the same design as another conducted in Pakistan and Brazil by Chaudhry and colleagues [37]. While the latter reported an improvement of negative symptoms in the group receiving minocycline, the former did not find a beneficial effect of minocycline in the progression and severity of symptoms in participants with a recent diagnosis. One other study [45] reported no beneficial effect of minocycline on any of the variables investigated. These differences might explain the high heterogeneity observed in our meta-analysis (I^2^ = 69% and I^2^ = 85% for total and negative symptoms, respectively).

Collectively, the studies included in this review support a significant effect of minocycline on negative symptoms of schizophrenia. One study [43] proposed that such effects are associated with a reduction in serum levels of pro-inflammatory cytokines (interleukin (IL)-1β, IL-6, and TNF-α). Notably, Liu and colleagues [39] reported no difference in the levels of IL-1β and TNF-α after 16 weeks of treatment but reported interesting correlations between nitric oxide (NO) and improvement of Scale for the Assessment of Negative Symptoms (SANS) score. Lowered levels of NO also correlated with worse symptoms. Although the operative mechanisms of minocycline in schizophrenia remain unclear, the literature available points towards an anti-inflammatory effect and modulation of neurotransmitters through the NMDA-NO-cGMP pathway [69].

#### 3.2.2. Major Depressive Disorder

Significant differences between groups for depressive symptoms in MDD favoring minocycline (SMD = −0.36 [−0.71 −0.01], I^2^ = 0%) were observed. No significant between-group difference for either anxiety symptoms (SMD = −0.33 [−0.75, 0.09], I^2^ = 0%) or quality of life (SMD = −0.23 [−0.65, 0.19], I^2^ = 0%), in Figure 3, were found.

Three papers studying the effects of minocycline on depression were included in the review. These included data for 75 participants that received minocycline at 200 mg per day for 4 to 16 weeks. Our meta-analysis, although possibly underpowered due to the small sample size, points towards the beneficial effect of minocycline when compared to the placebo, reinforcing the observation of the 3 studies included. Dean and colleagues [47] remark that although the four-point difference observed in their study was not significant, it is of the same magnitude of effect as other widely used antidepressants. More recently, Zazula et al. [70], combined the data from 2 of the studies included in this review (Dean [47] and Husain [48]) to conduct pooled analysis. They found a significant reduction in the symptoms of depression and anxiety at week 12, further supporting the effect of minocycline on depression. 

Analyses by Nettis et al. [49] suggested again that minocycline effects may be due to its anti-inflammatory properties, showing that participants with higher serum levels of C-reactive protein (CRP) and IL-6 at baseline had a greater reduction on HAM-D-17 scores. Changes in IL-6 levels over 12 weeks of adjunctive minocycline treatment were also associated with improved anxiety symptoms from the Dean et al. trial in MDD [71].

#### 3.2.3. Bipolar Disorder

No significant between-group differences for depressive symptoms in bipolar disorder were observed (SMD = 0.16 [−0.17, −0.46], I^2^ = 0%), in Figure 4, were found.

Two studies were included in this review, with a total of 85 participants who received minocycline at 200 mg per day for 6 and 12 weeks. These studies included minocycline and another drug of interest (celecoxib [50] and aspirin [51]), and both found no significant beneficial effect of minocycline on bipolar depression. Savitz et al. [51], however, observed that participants receiving both minocycline and aspirin were twice as likely to respond to treatment when compared to the placebo group. Participants who manifested inflammation (indexed by higher levels of IL-6 at baseline) responded better to minocycline than those without baseline inflammation. Similar to the section above, these results might be under the power of the small number of studies included.

#### 3.2.4. Substance Use

The meta-analysis did not suggest a significant effect of minocycline in any of the outcomes assessed including cognitive function (SMD = −0.26 [−0.82, 0.30]) and craving (SMD = −0.38 [−1.46, 0.71]) (Figure 5).

One opioid dependence and one smoking study were included for substance use disorders, where 22 participants received minocycline at 200 mg per day for 4–15 days. Both studies have different designs: Sofluoglu [53] used a crossover setting, with an adaptation session and two 4-day treatment periods when the outcomes were measured. There was no effect of minocycline in smoking self-administration as well as no differences between minocycline and the placebo on measures of behavior, physiological, and cognitive domains. Minocycline seemed to be associated with a modest improvement in depressed mood and a reduction of cravings. Arout and colleagues [52], on the other hand, used an individually randomized parallel-group setting that included adults enrolled in an opioid agonist treatment program. This study consisted of one adaptation session followed by three test sessions and one follow-up visit after treatment cessation. The authors report that minocycline treatment was associated with a more accurate performance on the Go/No-go task (measuring impulsivity), but had no effect on pain threshold and tolerance, withdrawal, or craving. Here again, the comparison might be under the power of the inclusion of only 2 studies.

#### 3.2.5. OCD

One study [54] showed that minocycline adjunctive to fluvoxamine reduced the OCD symptoms measured by the Yale-Brown Obsessive Compulsive Scale (Y-BOCS) in participants with moderate-to-severe OCD.

### 3.3. Neurological Disorders

#### 3.3.1. Stroke

When considering the NIHSS at day 90 post-stroke, the evidence did not suggest a benefit of minocycline (SMD −0.58 [−1.21, 0.05]; I^2^ = 87%; *p* = 0.07), and, likewise, on the Barthel Index (SMD 0.47 [−0.16, 1.09]; I^2^ = 85%, *p* = 0.14). However, a significant benefit of minocycline on mean mRS score (SMD −0.71 [−1.20, −0.22]; I^2^ = 54%; *p* = 0.004), but not on category change to independent functioning (RR 1.36 [0.97, 1.90], I^2^ = 74%, *p* = 0.08), was observed (Figure 6). We observed no evidence of publication bias (Appendix A).

Seven studies investigated the use of minocycline for stroke (ischemic event or hemorrhagic), including 533 participants, with 263 in the minocycline group (doses from 100 mg to 400 mg per day) and 270 in the control group, for 5 days. The control groups included a placebo, routine care, treatment as usual, and vitamin D. Four of the 7 studies [55,56,58,59] reported a significant effect of minocycline on the outcomes measured and included participants that had an ischemic stroke only, while the remaining three [57,60,61] reported no significant differences between minocycline treatment and control groups, and included participants with ischemic, hemorrhagic, and both.

#### 3.3.2. Injury

Two studies [62,63] included in this review reported the effects of minocycline on brain and spinal injuries. These studies included 86 participants, of which 41 received minocycline at 200 mg–800 mg per day for 7 days. Both studies suggest that minocycline was associated with an improvement in neurological and functional outcomes.

#### 3.3.3. Other

The search strategy found one study for each of the following conditions: amyotrophic lateral sclerosis (ALS) [64]—inconclusive study; Alzheimer’s disease (AD) [65]—no benefits associated with minocycline; Multiple system atrophy (MSA) [66]—no beneficial effect of minocycline; arthritis [67]—minocycline was effective in reducing joint swelling and tenderness at 12 weeks and improvement continued to increase through to week 48; pain [68]—no benefit with minocycline treatment.

#### 3.3.4. Limitations

This meta-analysis is not without limitations. First, the data extracted for analysis were limited to post-intervention scores without adjustment for pre-intervention values. While the inclusion criteria specified that all studies must be randomized control trials, random allocation to treatment conditions does not entirely eliminate non-equivalency [72]. We attempted to control for the influence of non-equivalency by including baseline outcome scores as a meta-regressive predictor, among others. Second, except for schizophrenia studies, the total number of included studies was low and the sample sizes were small. Underpowered studies are more likely to report statistically significant findings and drive greater between-study variability increasing heterogeneity [73]. As such, meta-analysis results that include smaller studies must be interpreted with greater caution even if the total pooled sample size is large.

## 4. Conclusions

There is limited data on the use of minocycline as an adjunctive treatment for psychiatric and neurological conditions in terms of quality and quantity of studies. Studies in psychiatry suggest an overall benefit of minocycline, especially in schizophrenia, where it seems to act through anti-inflammatory and neurotransmitter pathways to exert its effects. In the field of neurology, however, despite some positive effects, larger controlled trials are needed to reach definitive conclusions, including exploring possible mechanisms of action. Overall, minocycline appears safe and tolerable for both neurological and psychiatric conditions. More work is needed to improve evidence for definitive conclusions but minocycline also appears to be a promising candidate for the treatment of psychiatric disorders in particular.

## Figures and Tables

**Figure 1 ijms-24-05250-f001:**
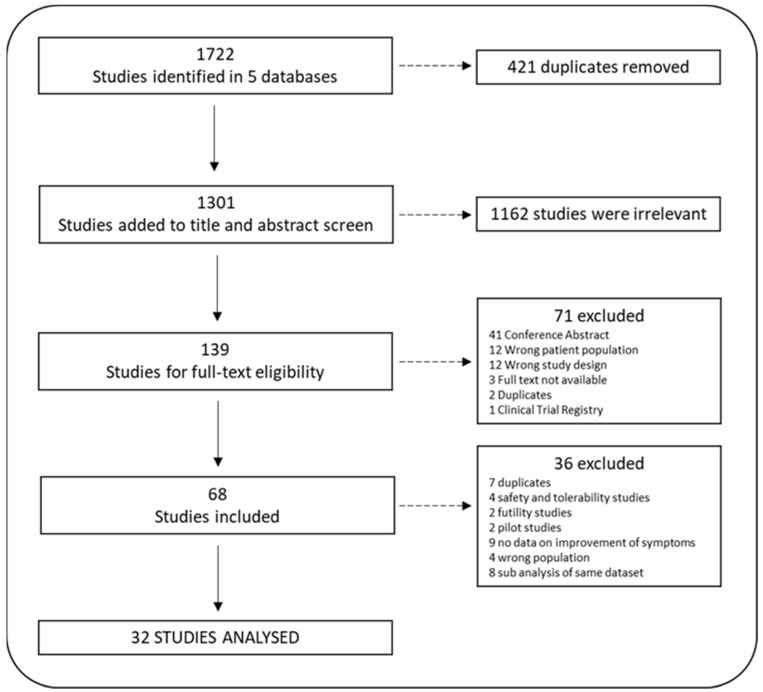
Prisma flowchart: systematic review steps.

**Figure 2 ijms-24-05250-f002:**
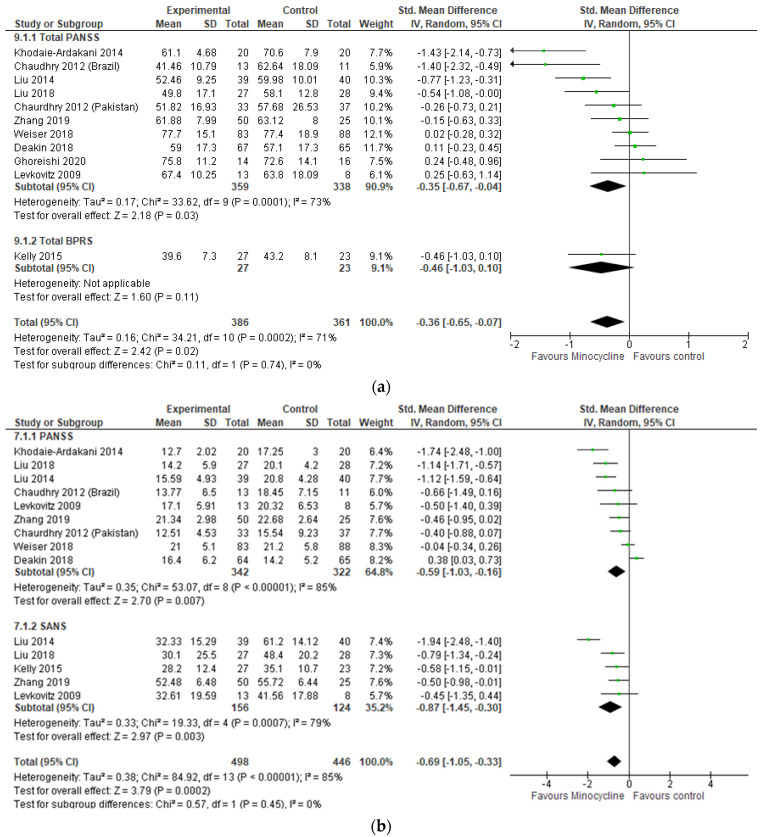
Meta-analysis results for schizophrenia. (**a**) Total symptoms; (**b**) Negative symptoms; (**c**) Positive symptoms; (**d**) General symptoms; (**e**) Extrapyramidal symptoms; (**f**) CGI; (**g**) GAF; (**h**) AIMS; (**i**) Depressive symptoms; and (**j**) Cognitive function: processing speed, working memory, psychomotor speed, executive functions, problem-solving, verbal learning, visual learning [37,38,39,40,41,42,43,44,45,46].

**Figure 3 ijms-24-05250-f003:**
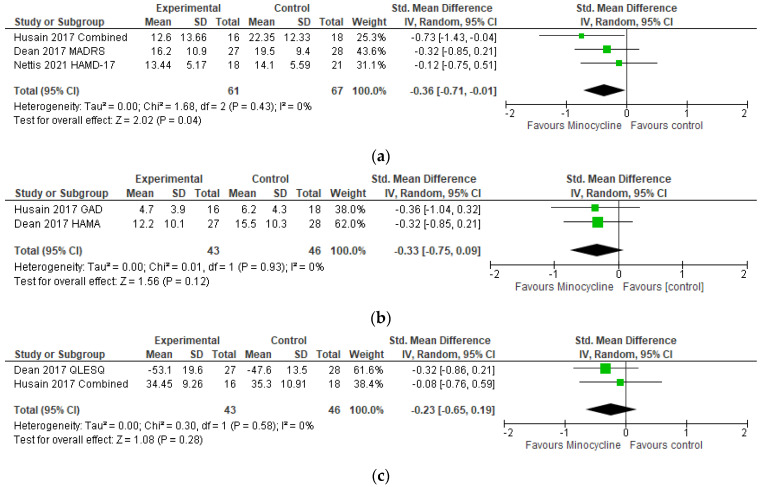
Meta-analysis results for MDD. (**a**) Depression; (**b**) Anxiety; (**c**) Quality of life [47,48,49].

**Figure 4 ijms-24-05250-f004:**
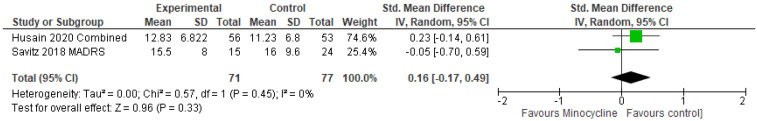
Meta-analysis results for BD [50,51].

**Figure 5 ijms-24-05250-f005:**
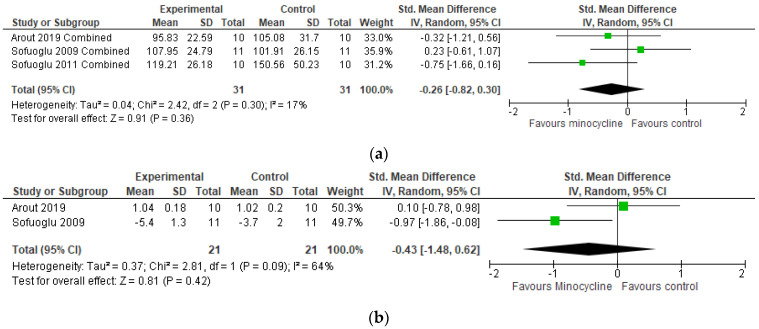
Meta-analysis results for substance use. (**a**) Cognitive function; (**b**) Craving [52,53].

**Figure 6 ijms-24-05250-f006:**
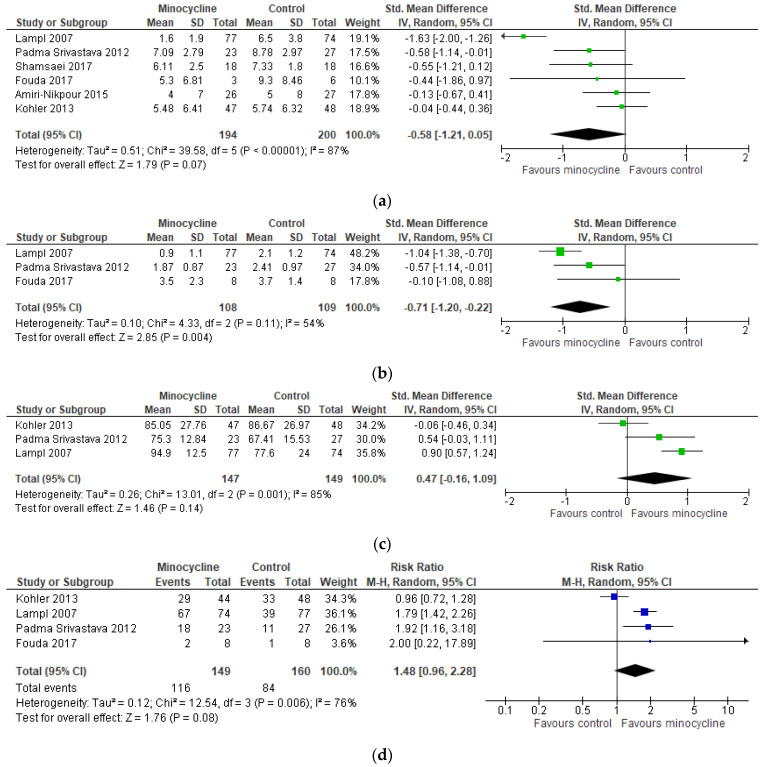
Meta-analysis results for stroke. (**a**) NIHSS; (**b**) MRS; (**c**) Barthel; (**d**) MRS category change [55,56,57,58,59,60,61].

**Table 1 ijms-24-05250-t001:** PubMed search strategy.

Search	Query
#1	minocycline[MeSH Terms]
#2	minocycline[Title/Abstract]
#3	#1 OR #2
#4	neurology [MeSH terms] OR Child Development Disorders, Pervasive [MeSH terms] OR dementia [MeSH terms] OR amyotrophic lateral sclerosis [MeSH terms] OR multiple sclerosis [MeSH terms] OR neurodegenerative diseases [MeSH terms] OR migraine disorders [MeSH terms] OR headache disorders [MeSH terms] OR brain injury, traumatic [MeSH terms] OR cerebrovascular disorders [MeSH terms] OR epilepsy [MeSH terms] OR seizures [MeSH terms] OR Ischemia [MeSH terms] OR psychiatry [MeSH terms] OR mental disorders [MeSH terms] OR anxiety [MeSH terms] OR Disruptive, Impulse Control, and Conduct Disorders [MeSH terms] OR nail biting [MeSH terms] OR cannabis [MeSH terms] OR benzodiazepines [MeSH terms] OR nicotine [MeSH terms] OR analgesics, opioid [MeSH terms] OR cocaine [MeSH terms] OR heroin [MeSH terms] OR methamphetamine [MeSH terms] OR amphetamine [MeSH terms] OR methylphenidate [MeSH terms] OR substance-related disorders [MeSH terms]
#5	(((((((((((((((((((((((((((((((((((((((((((((((((((((((((((((((((((((((((((((((((((((((((“neurological disorder”[Title/Abstract]) OR (neurology[Title/Abstract])) OR (autism[Title/Abstract])) OR (“autistic disorder”[Title/Abstract])) OR (ASD[Title/Abstract])) OR (Asperger’s[Title/Abstract])) OR (“Asperger Syndrome”[Title/Abstract])) OR (“pervasive developmental disorder”[Title/Abstract])) OR (“Alzheimer’s disease”[Title/Abstract])) OR (dementia[Title/Abstract])) OR (“amyotrophic lateral sclerosis”[Title/Abstract])) OR (ALS[Title/Abstract])) OR (seizures[Title/Abstract])) OR (epilepsy[Title/Abstract])) OR (“traumatic brain injury”[Title/Abstract])) OR (TB[Title/Abstract])) OR (stroke[Title/Abstract])) OR (ischemia[Title/Abstract])) OR (ischaemia[Title/Abstract])) OR (haemorrhage[Title/Abstract])) OR (neuropathy[Title/Abstract])) OR (“peripheral neuropathy”[Title/Abstract])) OR (neurodegenerative[Title/Abstract])) OR (“Parkinson disease”[Title/Abstract])) OR (“Huntington disease”[Title/Abstract])) OR (“brain injury”[Title/Abstract])) OR (“multiple sclerosis”[Title/Abstract])) OR (“spinal cord injury”[Title/Abstract])) OR (migraine[Title/Abstract])) OR (headache[Title/Abstract])) OR (“psychiatric disorder”[Title/Abstract])) OR (“mental illness”[Title/Abstract])) OR (“mental disorders”[Title/Abstract])) OR (anxiety[Title/Abstract])) OR (addiction[Title/Abstract])) OR (“major depressive disorder”[Title/Abstract])) OR (“major depression”[Title/Abstract])) OR (“depressive disorder”[Title/Abstract])) OR (MDD[Title/Abstract])) OR (“bipolar I disorder”[Title/Abstract])) OR (“bipolar II disorder”[Title/Abstract])) OR (“bipolar disorder”[Title/Abstract])) OR (“bipolar depression”[Title/Abstract])) OR (mania[Title/Abstract])) OR (“manic disorder”[Title/Abstract])) OR (“manic state”[Title/Abstract])) OR (hypo-mania[Title/Abstract])) OR (psychosis[Title/Abstract])) OR (“psychotic disorders”[Title/Abstract])) OR (schizophrenia[Title/Abstract])) OR (“schizoaffective disorder”[Title/Abstract])) OR (“panic disorder”[Title/Abstract])) OR (“social anxiety disorder”[Title/Abstract])) OR (PTSD[Title/Abstract])) OR (“posttraumatic stress disorder”[Title/Abstract])) OR (“generalised anxiety disorder”[Title/Abstract])) OR (OCD[Title/Abstract])) OR (“personality disorder”[Title/Abstract])) OR (“obsessive compulsive disorder”[Title/Abstract])) OR (“obsessive-compulsive neurosis”[Title/Abstract])) OR (“obsessive-compulsive neuroses”[Title/Abstract])) OR (“attention deficit hyperactivity disorder”[Title/Abstract])) OR (ADHD[Title/Abstract])) OR (trauma[Title/Abstract])) OR (“stress disorders”[Title/Abstract])) OR (“post traumatic”[Title/Abstract])) OR (“feeding disorder”[Title/Abstract])) OR (“appetite disorder”[Title/Abstract])) OR (disruptive, impulse control, conduct disorders[Title/Abstract])) OR (“eating disorder”[Title/Abstract])) OR (“binge eating disorder”[Title/Abstract])) OR (“bulimia nervosa”[Title/Abstract])) OR (“anorexia nervosa”[Title/Abstract])) OR “other specified feeding and eating disorder”[Title/Abstract])) OR (methylphenidate[Title/Abstract])) OR (amphetamine[Title/Abstract])) OR (methamphetamine[Title/Abstract])) OR (cocaine[Title/Abstract])) OR (cannabis[Title/Abstract])) OR (marijuana[Title/Abstract])) OR (heroin[Title/Abstract])) OR (“prescription pills”[Title/Abstract])) OR (opioids[Title/Abstract])) OR (benzodiazepine[Title/Abstract])) OR (nicotine[Title/Abstract])) OR (tobacco[Title/Abstract])) OR (“pathological gambling”[Title/Abstract])) OR (“skin picking”[Title/Abstract])) OR (“impulse control disorder”[Title/Abstract])) OR (kleptomania[Title/Abstract])
#6	#4 OR #5
#7	(randomized controlled trial [pt] OR controlled clinical trial [pt] OR randomized [tiab] OR placebo [tiab] OR clinical trials as topic [mesh: noexp] OR randomly [tiab] OR trial [ti]
#8	animals [mh] NOT humans [mh]
#9	#7 NOT #8
#10	#3 AND #6 AND #9
#11	#3 AND #6 AND #9 Filters: English; Humans

**Table 2 ijms-24-05250-t002:** Studies included in this systematic review.

	Author	Country	Disease	N Minocycline	N Control Group	Control Group (Placebo)	Primary Outcome	Treatment	Treatment Length	Conclusion	RoB
1	Chaundhry 2012 [37]	Pakistan and Brazil	SZ—within 5 years of onset	71	73	Y	PANSS	50 mg increments until 200 mg/daily	12 months	The addition of minocycline to TAU early in the illness course of schizophrenia improves negative symptoms without a detectable effect on cognition.	Some concerns
2	Liu 2014 [38]	China	SZ—early stage	39	40	Y	SANS/PANSS	200 mg/daily	16 weeks	The addition of minocycline to atypical antipsychotic drugs in early schizophrenia had significant efficacy on negative symptoms but had a slight effect on the attention domains of patients with schizophrenia.	High Risk
3	Liu 2018 [39]	China	SZ	27	28	Y	SANS/PANSS	200 mg/daily	16 weeks	16-week adjunctive minocycline treatment significantly improved schizophrenia symptoms, in particular the negative symptoms.	High Risk
4	Levkovitz 2010 [40]	Israel	SZ	13	8	Y	SANS/PANSS	200 mg/daily	24 weeks—2 weeks single-blind placebo lead-in phase	Minocycline treatment was associated with improvement in negative symptoms and executive function, both related to frontal lobe activity. Supports benefits of minocycline for schizophrenia.	High Risk
5	Khodaie-Ardakani 2014 [41]	Iran	SZ—with a duration of >2 years	20	18	Y	PANSS	100 mg/daily for 1 week then increase to 200 mg/daily	8 weeks	Minocycline seems to be beneficial and tolerable as a short-term add-on to risperidone for the treatment of negative and general psychopathology symptoms of schizophrenia.	Some concerns
6	Kelly 2015 [42]	USA	SZ	28	23	Y	BPRS/SANS	100 mg/daily increase to 200 mg/daily	13 weeks: 3 weeks screening and stabilization + 10 weeks treatment	Minocycline’s effect on the MCCB composite score and positive symptoms was not statistically significant. Significant improvements with minocycline were seen in working memory, avolition, and anxiety/depressive symptoms in a chronic population with persistent symptoms.	Low risk
7	Zhang 2018 [43]	China	SZ	25 LD and 25 HD	25	Y	PANSS/SANS	100 mg/daily = low dose; 200 mg/daily = high dose	3 months	The addition of high-dose minocycline to risperidone reduced the negative symptoms of patients with schizophrenia. The improvement in negative symptoms correlated with the reduction in serum levels of pro-inflammatory cytokines. Participants receiving high-dose minocycline had a greater reduction in SANS and PANSS negative subscale scores than those receiving low-dose minocycline or placebo.	High risk
8	Deakin 2018 [44]	UK	SZ—first episode	103	104	Y	PANSS	200 mg/daily for 2 weeks, then 300 mg/daily for the remainder of the 12-month period	12 months	12 months’ treatment with minocycline does not improve the symptomatic or functional status of people within 5 years of a diagnosis of schizophrenia.	Low risk
9	Ghoreishi 2020 [45]	Iran	SZ—<10 years diagnosis	14	16	Y	PANSS	200 mg/daily	8 weeks	Minocycline could correct positive and negative symptoms of schizophrenia.	High risk
10	Weiser 2018 [46]	Romania and the Republic of Moldova	SZ	100	100	Y	PANSS	200 mg/daily	16 weeks	Minocycline was not efficacious in treating negative symptoms or cognition.	High risk
11	Dean 2017 [47]	Australia and Thailand	MDD	36	35	Y	MADRS	200 mg/daily	16 weeks of treatment with a follow-up visit at week 16.	The four-point difference between minocycline and placebo in the MADRS scores (primary outcome measure) was not significant, this is on par with the magnitude of change seen with conventional antidepressants. There were significant improvements in several important outcomes including global impression, functioning, and quality of life.	Low risk
12	Husain 2017 [48]	Pakistan	MDD	21	20	Y	HAMD-16	100 mg/daily for 2 weeks then 200 mg/daily for 10 weeks	12 weeks	Minocycline has a potential role as an augmentation strategy in patients with treatment-resistant depression.	Some concerns
13	Nettis 2021 [49]	England	MDD	18	21	Y	HAMD-17	200 mg/daily	4 weeks	Participants selected for elevated C-reactive protein (≥1 mg/L) and found no clear difference between minocycline and placebo in improving depressive symptoms at week 4.	Some concerns
14	Husain 2020 [50]	Pakistan	BD I and II on the current MD episode	66	66	Y, with extra arms for celecoxib and Mino + Calecoxib	HAMD-24	100 mg/daily for 2 weeks then 200 mg/daily	12 weeks	Neither minocycline nor celecoxib were associated with an antidepressant effect for bipolar depression.	Low risk
15	Savitz 2018 [51]	USA	BD I, II and NOS	19	20	Y, with extra arms for Apirin and Mino + Aspirin	MADRS	200 mg/daily	6 weeks	No significant main effect of aspirin or minocycline on the mean change in MADRS score across visits. Participants receiving minocycline plus aspirin are twice as likely to show sustained response (OR = 2.93).	High risk
16	Arout 2019 [52]	USA	Opioid dependence	10	10	Y	CTP and BPI-SF	200 mg/daily	15 days	Neither pain threshold nor tolerance for pain in the CPT was affected by minocycline treatment. There was no evidence that minocycline had an effect on BPI, SOWS, POMS, or McGill measures assessed in the lab, or pain, craving, and opioid withdrawal assessed during EMA.	High risk
17	Sofuoglu 2009 [53]	USA	Smoking	12	-	Y	Choice procedure	200 mg/daily	4 days	There was no treatment effect on smoking self-administration. No differences between minocycline and placebo on most of the measures of behavioral, biochemical, physiological, subjective, and cognitive domains.	High risk
18	Esalatmanesh 2016 [54]	Iran	OCD	47	47	Y	Y-BOCS	200 mg/daily	10 weeks	Minocycline significantly reduced OCD symptoms as an adjuvant agent to fluvoxamine in moderate-to-severe OCD patients compared to a placebo.	High risk
19	Lampl 2007 [55]	Israel	Acute ischemic stroke	74	77	Y	NIHSS/BI/mRS	200 mg/daily	5 days	Patients with acute stroke had a significantly better outcome with minocycline treatment compared with a placebo in all outcomes.	High risk
20	PadmaSrivatava 2012 [56]	India	Acute ischemic stroke	23	27	Vit B	NIHSS/BI/mRS	200 mg/daily	5 days	Minocycline was observed to be beneficial in patients with acute ischemic stroke and was associated with better clinical and functional outcomes.	High risk
21	Kohler 2013 [57]	Australia	Acute stroke (ischemic or hemorrhagic)	47	48	Routine care	NIHSS/mRS	100 mg/daily	12 hourly for 5 doses	In this pilot study of a small sample of acute stroke patients, intravenous minocycline was safe but not efficacious. The study was not powered to identify reliably or exclude a modest but clinically important treatment effect of minocycline.	High risk
22	Amiri-Nikpour 2015 [58]	Iran	Acute ischemic stroke	26	27	Routine care	NIHSS	200 mg/daily	5 days	Patients who received minocycline had significantly better neurological outcomes on day 90 than controls. However, female patients showed no significant clinical improvement compared with males.	High risk
23	Shamsaei 2017 [59]	Iran	Acute ischemic stroke	18	18	Y	NIHSS	200 mg/daily	5 days	Both groups improved significantly over time, but neither group demonstrated superiority at 90-day follow-ups.	High risk
24	Singh 2019 [60]	Singapore	Acute ischemic stroke	69	70	Y	NIHSS/BI/mRS	200 mg/daily	5 days	At day 90, no significant difference was noted in the proportion of subjects given minocycline for any of the outcomes.	Low risk
25	Fouda 2017 [61]	USA	Acute ICH	8	8	TAU	mRS	400 mg/daily	5 days	No significant differences between groups reported at day 90.	High risk
26	Koulaeinejad 2019 [62]	Iran	TBI (moderate to severe)	14	20		GCS	200 mg/daily	7 days	Conclude as a result of this pilot study that minocycline treatment was associated with improvement in neurological outcomes of acute TBI compared with a placebo, warranting further clinical trials.	High risk
27	Casha 2012 [63]	Canada	SCI—acute traumatic	27	25	Y	ASIA	200 mg/daily, increased to 800 mg	7 days	In a randomized, double-blind manner, the treatment was associated with an apparent improvement in neurological and functional outcomes compared with a placebo, warranting further formal investigations of efficacy.	Some concerns
28	Pontieri 2005 [64]	Italy	ALS	10	10	TAU	ALS-FRS	100 mg/daily	6 months	The study was not powered to assess the efficacy of treatment, thus, the lack of effect of combined riluzole/minocycline treatment on ALS-FRS score or pulmonary score is not negative for efficacy, but inconclusive.	High risk
29	Howard 2020 [65]	UK	AD (SMMSE > 23)	HD: 184; LD: 181	179	Y	sMMSE and BADLS	400 md or 200 mg/daily	24 months	Minocycline did not delay the progress of cognitive or functional impairment in people with mild AD during a 2-year period. This study also found that 400 mg minocycline is poorly tolerated in this population.	Low risk
30	Dodel 2010 [66]	Germany and Austria	MSA—Parkinsonian symptoms	32	31	Y	UMSARS/UPDRSII	100 mg/daily	48 weeks	No beneficial effect of minocycline, either symptomatic or neuroprotective. Treatment with minocycline failed to improve global ratings of motor function, such as the UMSARSII (primary endpoint) and UPDRSIII (secondary endpoint) over the 48-week observation period.	High risk
31	Tilley 1995 [67]	USA	Rheumatoid Arthritis	109	110	Y	MHAQ	200 mg/daily	48 weeks	The MIRA trial showed that minocycline is both effective and safe for treating patients with mild to moderately active rheumatoid arthritis. Benefit became evident after 12 weeks of therapy, and the proportion of patients treated with minocycline showing improvement continued to increase through week 48 of the study.	Low risk
32	Curtin 2017 [68]	USA	Carpal tunnel syndrome (pain)	66	65	Cross-over	The Brief Pain Inventory	200 mg 2 h prior to procedure, then 100 mg twice a day (200 mg/day) for 5 days	5 days	This is a negative study with the principal result failing to identify a clinically meaningful overall change in TPR following treatment with minocycline (used at common antibiotic doses) during the perioperative period surrounding CTR surgery and TFR surgery.	Low risk

Abbreviations: AD—Alzheimer’s disease, ALS—amyotrophic lateral sclerosis, BD—bipolar disorder, BPI-SF—Brief Pain Inventory-Short Form, BPRS—Brief Psychiatric Rating Scale, BADLS—Bristol Activities of Daily Life Scale, CPT—Continuous Performance Test, CTR—carpal tunnel release, HAM-D—Hamilton Depression Rating Scale, HD—high dose, ICH—intercranial hemorrhage, LD—low dose, MADRS—Montgomery Asberg Depression Rating Scale, MMSE—Mini Mental State Exam, MCCB—MATRIX Consensus Cognitive Battery, MHAQ—Modified Health Assessment Questionnaire, MDD—major depressive disorder, mRS—magnetic resonance spectroscopy, NIHSS—NIH Stroke Scale, NOS—not otherwise specified, OCD—obsessive-compulsive disorder, PANSS—Positive and Negative Symptoms Scale, POMS—Profile of Mood States, SANS—Scale for the Assessment of Negative Symptoms, SCI—silent cerebral infarction, SOWS—Subjective Opioid Withdrawal Scale, SZ—schizophrenia, TAU—treatment as usual, TBI—traumatic brain injury, TFR—trigger finger release, TRP—time to pain resolution, UMSARS—Unified Multiple System Atrophy Rating Scale, UPDRSII—Unified Parkinson’s Disease Rating Scale, RoB—Risk of Bias.

## Data Availability

Details about the meta-analysis can be obtained by request to the authors.

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
