# Peer review of "Minocycline as Treatment for Psychiatric and Neurological Conditions: A Systematic Review and Meta-Analysis"

_ijms, 2023, doi:10.3390/ijms24065250_

Round 1

Reviewer 1 Report

In their manuscript, B.Panizzutti and her colleagues presented a wide up-to-date systematic review and meta-analysis of the data available on Minocycline trials in the therapy of major psychiatric and neurological conditions as mono- or adjuvant therapy remedy. Prepared in accordance with PRISMA guidance, the meta-review is well organized and scientifically sound enough. This seems a good contribution to the field and should be accepted, in my opinion. The only concern is the reason why the authors excluded data on autism, one of the most interesting condition? On page 4, Table 1 contains the search inquiry the authors applied, and in includes "autism" and "autistic disorder". But there has bee nothing said about the autistic spectrum disorders. At the same time, my brief search in Pubmed  found at least two trials of Minocycline in autism: https://pubmed.ncbi.nlm.nih.gov/23566357 and https://pubmed.ncbi.nlm.nih.gov/27128958. I think the authors should either include these disorders or explain why they have reported nothing of them, while having used the terms "autism" and "autistic disorder" in the search. So, I regard this option as "minor revisions".

Author Response

Thank you for your time and feedback.

While we appreciate the importance and interest in Autism and our search indeed included 2 studies investigating the use of minocycline for autism (the same 2 pointed by the reviewer). Due to the inclusion and exclusion criteria defined for the review in the protocol only studies with population of 18 years or old were included. The two clinical trials mentioned have a mean age of 7.5 and 7.1 years, and therefore, were excluded. We clarified the inclusion and exclusion criteria on the manuscript. [2.2 Study selection_ lines 6-8].

Reviewer 2 Report

This submission reports a thorough systematic review with meta-analyses on minocycline as an adjunctive treatment to standard pharmacotherapy in several psychiatric and neurological conditions.

The article is very detailed and methodologically sound, and the results report some interesting findings. However, there are some issues that limit the understandability and the contextualization of this piece of research. In view of this, I have some suggestion that can hopefully increase the scientific relevance of this article:

1.     For readers not acquainted with the concept of drug repurposing and its application in neuropsychiatric conditions, I suggest that the Authors provide some background information in their Introduction section. Please refer to Fava’s eminent editorial published in World Psychiatry [https://doi.org/10.1002/wps.20481].

2.     Despite the appreciable level of detail of the Results section, the amount of data synthesized by this review requires a proper Discussion section.

2a.     A summary and interpretation of findings (or the significant ones at least) should be provided (in this regard, the statement at page 21, lines 50–53 is not a result but a hint for discussion and should thus be moved to the relevant section).

2b.     Notwithstanding the rationale supporting the potential utility of anti-inflammatory agents for psychiatric disorders [Miller and Raison, 2015https://doi.org/10.1001%2Fjamapsychiatry.2015.22], it has been recently shown that evidence supporting their use in the treatment of severe mental disorders is weak and their clinical perspectives remain uncertain. In this regard, the Authors should refer to a recent umbrella review on the topic by Bartoli et al. [2021, https://doi.org/10.1016/j.jpsychires.2021.09.018]. The Authors should discuss their findings in this perspective, considering that, as for many other repurposed drugs, current evidence is limited – with most meta-analyses being underpowered and only few of them having significant results – and does not allow making reliable recommendations for the use of minocycline in clinical practice.

2c.     Limitations: the article completely lacks a limitations paragraph, which should be added before the conclusive remarks.

3.     The closing sentence of the manuscript seems a bit overconfident considering the actual findings of this work,. It should be rephrased in a more conservative way.

Author Response

Thank you for your time and feedback regarding our manuscript. We have addressed the comments below.

  1. For readers not acquainted with the concept of drug repurposing and its application in neuropsychiatric conditions, I suggest that the Authors provide some background information in their Introduction section. Please refer to Fava’s eminent editorial published in World Psychiatry [https://doi.org/10.1002/wps.20481].

Thank you for the suggestion. Additional text was added to the manuscript to clarify the repurposing concept and the reference suggested was also added. [1. Introduction lines 43-48]

  1. Despite the appreciable level of detail of the Results section, the amount of data synthesized by this review requires a proper Discussion section.

We agreed with the reviewer originally the sections were separated, but the journal guidelines are to have results and discussion sections together as presented in the manuscript.

2a.     A summary and interpretation of findings (or the significant ones at least) should be provided (in this regard, the statement at page 21, lines 50–53 is not a result but a hint for discussion and should thus be moved to the relevant section).

 We were instructed by the editorial team of the journal to keep results and discussion together. Therefore, we believe that the sections highlighted is in the relevant section.

2b.     Notwithstanding the rationale supporting the potential utility of anti-inflammatory agents for psychiatric disorders [Miller and Raison, 2015, https://doi.org/10.1001%2Fjamapsychiatry.2015.22], it has been recently shown that evidence supporting their use in the treatment of severe mental disorders is weak and their clinical perspectives remain uncertain. In this regard, the Authors should refer to a recent umbrella review on the topic by Bartoli et al. [2021, https://doi.org/10.1016/j.jpsychires.2021.09.018]. The Authors should discuss their findings in this perspective, considering that, as for many other repurposed drugs, current evidence is limited – with most meta-analyses being underpowered and only few of them having significant results – and does not allow making reliable recommendations for the use of minocycline in clinical practice.

The authors appreciate that there has been varying efficacy from many of the adjunctive anti-inflammatory trials.  Many of these trials have indeed been negative or have failed to separate significantly from placebo.  The current review aimed to try and address that heterogeneity in regards to minocycline specifically, by conducting a systematic evaluation of the literature. The authors appreciate that the broader context regarding anti-inflammatory treatments for psychiatric disorders is limited and have included the following in the Introduction:

This review is also timely in the context of adjunctive anti-inflammatory treatments where there is often heterogeneity in the literature regarding the field broadly and specific agents being trialled. [1. Introduction lines_ 53-56]

Results and Discussion: limitations and comments about the data collected are discussed throughout this section.

The first line in our conclusion states: ‘There is limited data on the use of minocycline as adjunctive treatment for psychiatric and neurological conditions in terms of quality and quantity of studies.’ [4. Conclusion_  lines 173-175]

2c.     Limitations: the article completely lacks a limitations paragraph, which should be added before the conclusive remarks.

The below text was added as limitations. [3.4.9 Limitations _ lines 158-171].

3.4.9 Limitations

                The meta-analysis is not without limitation. First, the data extracted for analysis were limited to post-intervention scores without adjustment for pre-intervention values. While the inclusion criteria specified that all studies must be randomized control trials, random allocation to treatment conditions does not entirely eliminate non-equivalency [71]. We attempted to control for the influence of non-equivalency by including baseline outcome scores as a meta-regressive predictor, among others. Second, except for schizophrenia studies, the total number of included studies was low and the sample sizes were small. Underpowered studies are more likely to report statistically significant findings and drive greater between-study variability increasing heterogeneity [72]. As such, meta-analysis results that include smaller studies must be interpreted with greater caution even if the total pooled sample size is large.

  1. Steeger, C.M.; Buckley, P.R.; Pampel, F.C.; Gust, C.J.; Hill, K.G. Common Methodological Problems in Randomized Controlled Trials of Preventive Interventions. Prev Sci 2021, 22, 1159–1172, doi:10.1007/s11121-021-01263-2.
  2. Turner, R.M.; Bird, S.M.; Higgins, J.P.T. The Impact of Study Size on Meta-Analyses: Examination of Underpowered Studies in Cochrane Reviews. PLOS ONE 2013, 8, e59202, doi:10.1371/journal.pone.0059202.

  1. The closing sentence of the manuscript seems a bit overconfident considering the actual findings of this work. It should be rephrased in a more conservative way.

The closing sentence was changed to: “Overall, minocycline appears safe and tolerable for both neurological and psychiatric conditions. More work is needed to improve evidence for definitive conclusions but minocycline also appears to be promising for the treatment of psychiatric disorders in particular.” [4. Conclusion _lines 180-183]

Round 2

Reviewer 2 Report

I appreciate the newly-added statements. However, the one at lines 52-54 of the Introduction section needs a proper reference, as suggested in my previous review report.

Author Response

Thank you for your comment and feedback. As suggested 2 critical references were added to the sentence in Introduction lines 51-53.

  1. Dean, O.M.; Data-Franco, J.; Giorlando, F.; Berk, M. Minocycline: Therapeutic Potential in Psychiatry. CNS Drugs 2012, 26, 391–401, doi:10.2165/11632000-000000000-00000.
  2. Berk, M.; Vieta, E.; Dean, O.M. Anti-Inflammatory Treatment of Bipolar Depression: Promise and Disappointment. Lancet Psychiatry 2020, 7, 467–468, doi:10.1016/S2215-0366(20)30155-3.